# Towards ultrafast dynamics with split-pulse X-ray photon correlation spectroscopy at free electron laser sources

W. Roseker[1], S.O. Hruszkewycz[2], F. Lehmkühler [1,3], M. Walther[1], H. Schulte-Schrepping[1], S. Lee[4,5], T. Osaka[6,10], L. Strüder[7], R. Hartmann [7], M. Sikorski[8,11], S. Song [8], A. Robert [8], P.H. Fuoss[2,12], M. Sutton[9], G.B. Stephenson[2] & G. Grübel[1,3]

One of the important challenges in condensed matter science is to understand ultrafast, atomic-scale fluctuations that dictate dynamic processes in equilibrium and non-equilibrium materials. Here, we report an important step towards reaching that goal by using a state-of-the-art perfect crystal based split-and-delay system, capable of splitting individual X-ray pulses and introducing femtosecond to nanosecond time delays. We show the results of an ultrafast hard X-ray photon correlation spectroscopy experiment at LCLS where split X-ray pulses were used to measure the dynamics of gold nanoparticles suspended in hexane. We show how reliable speckle contrast values can be extracted even from very low intensity free electron laser (FEL) speckle patterns by applying maximum likelihood fitting, thus demonstrating the potential of a split-and-delay approach for dynamics measurements at FEL sources. This will enable the characterization of equilibrium and, importantly also reversible non-equilibrium processes in atomically disordered materials.

[1] Deutsches Elektronen-Synchrotron DESY, Notkestr. 85, 22607 Hamburg, Germany. [2] Materials Science Division, Argonne National Laboratory, Argonne, IL 60439, USA. [3] The Hamburg Centre for Ultrafast Imaging, Luruper Chaussee 149, 22761 Hamburg, Germany. [4] Frontier in Extreme Physics, Korea Research Institute of Standards and Science, Daejeon 305-340, Korea. [5] Department of Nanoscience, University of Science and Technology, Daejeon 305-350, Korea. [6] Department of Precision Science and Technology, Graduate School of Engineering, Osaka University, 2-1 Yamada-oka, Suita, Osaka 565-0871, Japan. [7] PNSensor GmbH, Otto-Hahn-Ring 6, 81739 München, Germany. [8] Linac Coherent Light Source, SLAC National Accelerator Laboratory, Menlo Park, CA 94025, USA. [9] Department of Physics, McGill University, Montreal, Quebec H3A2T8, Canada. [10] Present address: RIKEN SPring-8 Center, 1-1-1 Kouto, Sayo-cho, Sayo-gun, Hyogo 679-5148, Japan. [11] Present address: European X-Ray Free-Electron Laser Facility, Holzkoppel 4, 22869 Schenefeld, Germany. [12] Present address: SLAC National Accelerator Laboratory, Menlo Park, CA 94025, USA. Correspondence and requests for materials should be addressed to W.R. (email: wojciech.roseker@desy.de)

X-ray photon correlation spectroscopy[1] (XPCS) is a powerful tool to study slow dynamics in complex systems and is routinely used at storage ring sources on time scales of milliseconds to hours. Applied with ultrashort coherent X-ray pulses available at current free electron laser (FEL) sources[2–4], it can potentially track atomic scale fluctuations[5–7] in liquid metals[8], multi-scale dynamics in water[9], fluctuations in the undercooled state[10–12], heterogeneous dynamics about the glass transition[13–15], and atomic scale surface fluctuations[16,17]. In addition, time-domain XPCS[18,19] at FEL sources is well suited for studying fluctuations in reversible non-equilibrium processes that go beyond time-averaged structural descriptions. Such pump (split-pulse) probe experiments are feasible for non-equilibrium processes that are reversible on the timescale defined by the repetition rate of the experiment. This will allow the elucidation of dynamics of ultrafast magnetization processes[20–22] and can address open questions concerning photo-induced phonon dynamics[23,24] and phase transitions[25].

We have used split-pulse XPCS to enable sub-ns dynamics to be measured at hard X-ray FEL sources that operate mostly with pulse spacings in the 10 ms regime[2,3]. This approach relies on diffractive optics[26–29] capable of splitting individual FEL pulses and introducing a tunable time delay $\Delta_t$ between the two sub-pulses that both diffract from the sample into a detector and produce a single speckle pattern. Analyzing the contrast of such patterns can give access to the underlying sample dynamics since dynamic processes with time constants longer than $\Delta_t$ will not influence the contrast, while processes faster than $\Delta_t$ will lead to a decorrelation of the speckle pattern and thus decrease its contrast. Sample dynamics are then probed by measuring split-pulse speckle contrast $\beta(q, \Delta_t)$ as a function of scattering wave vector $q$[30] and pulse separation. This approach can span timescales from tens of femtoseconds (the pulse width of the FEL) to several nanoseconds, well beyond the time resolution of any X-ray area detector.

Here, we addressed two key issues critical to this approach. First, reliable contrast values can be extracted from single split-pulse pattern only in very exceptional cases[31,32]. Since averaging of single split-pulse speckle patterns would eliminate the contrast information, we developed a method for reliably extracting contrast values individually from thousands of split-pulse patterns of low photon content from weakly scattering disordered systems. Second, we successfully accounted for the fact that the split-pulse contrast depends on the degree of decoherence (for example geometrical overlap) and splitting ratio between the two beams, both of which will change at today's FEL sources on a shot-to-shot basis.

In this work, we have demonstrated the feasibility of the split-pulse approach by measuring nanosecond dynamics of nanoparticles in suspension. This was accomplished by first calibrating the hard X-ray split-and-delay system at the XCS beamline at the Linac Coherent Light Source (LCLS)[33] by determining split-pulse contrast from a static sample. The device was tuned to a pulse delay of 1.3 ns, and split-pulse speckle patterns were obtained from a suspension of 1-nm-radius gold nanoparticles over a range of scattering vectors $q$. This sample was chosen such that the correlation times $\tau_c = 1/(D_0 q^2)$ describing the Brownian motion of the gold particles could be easily tuned to values below and above the selected instrumental delay time $\Delta_t$ by changing the momentum transfer $q$. This allowed us to map out the contrast $\beta(q, \Delta_t)$ as a function of $q$ for a given $\Delta_t$ and thus determine the diffusion coefficient. Our results confirm the expected dynamics of the system, and demonstrate the successful application of hard X-ray split-and-delay XPCS at free electron laser sources.

## Results

**Static split-pulse speckle contrast**. The split-pulse XPCS experiment was carried out using the setup shown in Fig. 1. A static film of 150-nm-radius silica particles was used to determine the baseline contrast in terms of the pulse intensity splitting ratio $r_{sp}$ and the degree of decoherence $\sigma_d$ of the two pulses at the sample (see Supplementary Note 3). Split-pulse data were collected with a pnCCD area detector[34]. The average count rates in a single split-pulse frame were in the range of 0.002–0.004 photons/pixel/pulse. A strong split-pulse speckle pattern from the static sample (after processing with photon fitting (see Supplementary Note 1)) is shown in Fig. 2a. A region of interest from this pattern is shown in Fig. 2c highlighting the low probability that a given pixel will contain single, double or triple photon hits. Despite the low count rate of any single split-pulse speckle pattern, the sum and azimuthally averaged mean intensity profile of thousands of such scattering patterns (Fig. 2b, d) show the average scattering behavior expected from the spherical particles.

The contrast of a speckle pattern $\beta$ can be expressed in terms of the number of coherent modes[35] $M = 1/\beta$. For a given photon count rate in the detector ($\lambda$, photons/pixel) and a number of modes $M$, the probability of observing $i$ number of photons in a

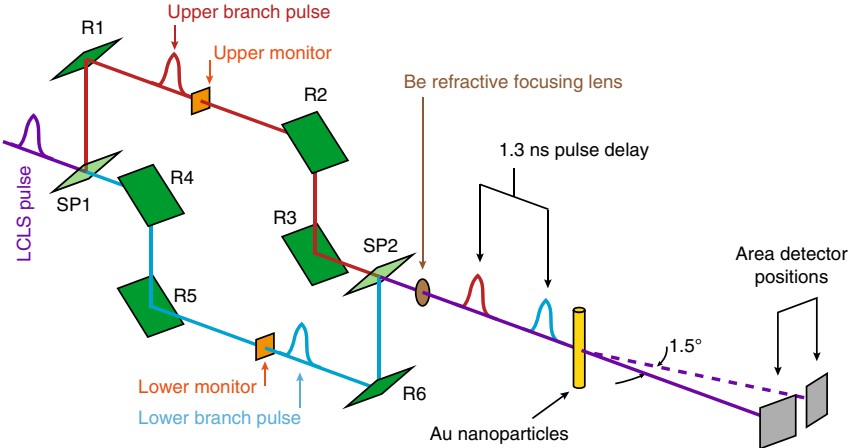

**Fig. 1** Schematics of the hard X-ray delay line instrument. An incoming ultrashort X-ray pulse is split with a thin silicon single crystal splitter (SP1). The pulses are directed along different pathways via reflective Bragg optics (R1–R6). The relative path length difference between the lower branch (LB) and the upper branch (UB) introduces a time delay $\Delta_t$ (here 1.3 ns) between the split pulses

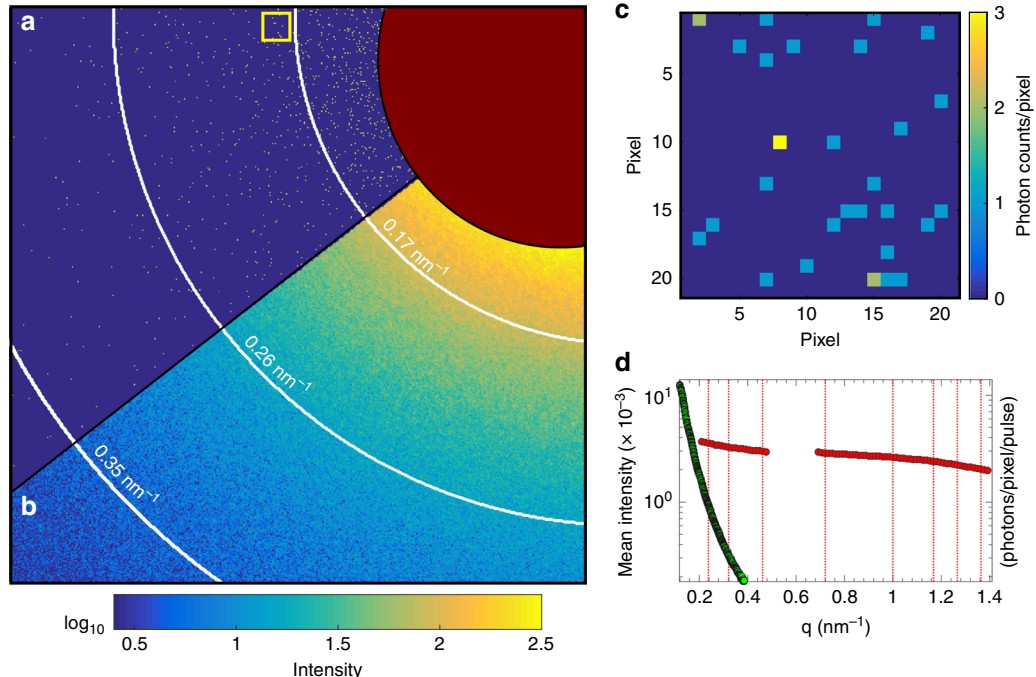

**Fig. 2** 2D speckle pattern. **a** Single split-pulse scattering pattern. **b** Sum of $2.5 \times 10^3$ scattering patterns. The brown circular area corresponds to the beamstop, which was masked in the data analysis process. **c** Signal of single (blue), double (green) and triple (yellow) photon events in a 22 by 22 pixel region of interest (shown by the yellow square in **a**). **d** Azimuthally integrated intensity of the summed scattering patterns collected from the static sample (green). Red circles denote the integrated intensity collected from the dynamic gold sample at two detector positions

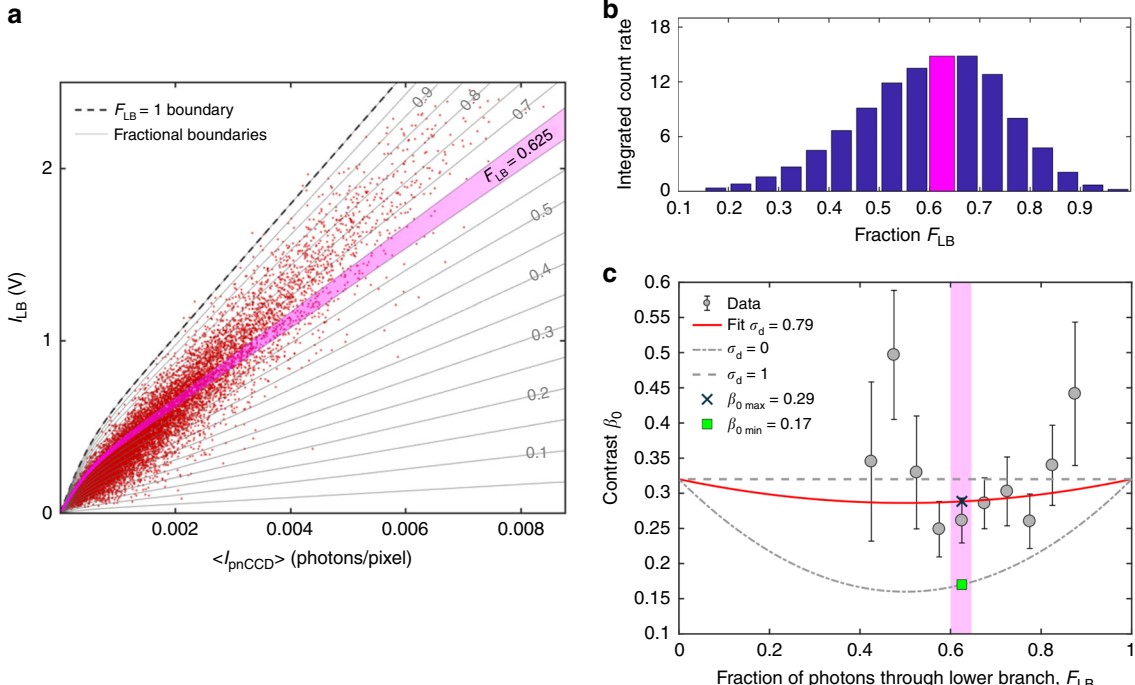

**Fig. 3** Static split-pulse speckle contrast. **a** Fraction $F_{LB}$ of photons passing through the lower branch derived from the lower branch intensity monitor signal $I_{LB}(V)$ as a function of the pnCCD signal ($I_{pnCCD}$) for $1.5 \times 10^4$ FEL pulses. The non-linear form of the boundary $F_{LB} = 1$ is due to the optical setup configuration and the spiky nature of the FEL spectrum (see Supplementary Note 2). **b** Histogram of the integrated count rates (see Supplementary Fig. 5) at selected $q = 0.157$ nm$^{-1}$ as a function of $F_{LB}$. **c** Contrast observed for the static sample as a function of the $F_{LB}$. The best fit (red line) of Eq. (3) to the data yields a decoherence factor $\sigma_d$ of $0.79 \pm 0.35$. The limiting cases for perfectly aligned ($\sigma_d = 1$) and fully decoherent ($\sigma_d = 0$) are also shown by dashed and dashed-dotted lines, respectively. The magenta bar represents the contrast values for fraction $F_{LB} = 0.625 \pm 0.025$. The expected limiting contrast values are 0.29 and 0.17 as shown by blue cross and green square respectively. The error of data points was calculated based on Supplementary Eq. (3)

pixel is given by the negative binomial probability distribution

$$P_i(\lambda, M) = \frac{\Gamma(i+M)}{\Gamma(M)\Gamma(i+1)}\left(1+\frac{M}{\lambda}\right)^{-i}\left(1+\frac{\lambda}{M}\right)^{-M} \quad (1)$$

where $\Gamma$ is the gamma function. For low-intensity speckle patterns, such as shown in Fig. 2c, the number of pixels in the detector containing $i = 0, 1, 2,$ or 3 photons in a pixel ($n_i = n_0, n_1, n_2, n_3$) determine the contrast $\beta$[8]. The observed mean count rate $\hat{\lambda}_j$ (taken as an estimate of the true count rate $\lambda_j$) in the detector for each split-pulse pattern with a given splitting ratio (denoted by the index $j$) can be used to reduce Eq. (1) to a function of a single variable $M$ that can be estimated from a large number of speckle patterns by maximum likelihood fitting. This approach allows photon statistics from patterns with very low count rates to be meaningfully integrated into an estimate of $M$ of the entire set (see Supplementary Note 1).

Implicit in this fitting approach is the assumption that the contrast of all split-pulse patterns in the set have the same contrast. However, the contrast in a split-pulse speckle pattern from a sample depends on the splitting ratio $r_{sp}$ which can be affected by spectral and positioning fluctuations of the FEL beam[36–38], as well as the level of decoherence $\sigma_d$ between the two split pulses arising from imperfect area overlap, differences in incident angle and differences in wavelength (see Supplementary Note 3). It is thus critical to monitor the splitting ratio $r_{sp}$ shot by shot especially since significant deviations from the nominal 1:1 ratio were observed that need to be accounted for in our approach. In this experiment the splitting ratio was quantified using the monitor $I_{LB}(V)$ placed in the lower branch of the device in combination with the pnCCD detector (see Supplementary Note 2). Figure 3a shows the correlation between the average count rate in the pnCCD area detector and the LB (lower branch) monitor signal for $1.5 \times 10^4$ sequentially recorded split-pulse frames. For any split-pulse pattern with a given monitor signal $I_{LB}(V)$ one observes a band of pnCCD signals $\langle I_{pnCCD} \rangle$ due to the (varying) contribution of the upper branch to the pnCCD signal. The width of the pnCCD signal band is determined by fluctuations of the incident intensity, pointing stability and in particular splitting-ratio fluctuations introduced by spectral fluctuations of the incident beam. The frames with the lowest pnCCD count rate for a given monitor signal $I_{LB}(V)$, described by the dashed line, correspond to the cases where the contribution of the upper branch is zero and thus the fraction of intensity passing through the lower branch $F_{LB} = 1$. Based on the functional form of the $F_{LB} = 1$ profile (see Supplementary Note 2) we were able to establish contours on the correlation diagrams that delineate populations of split-pulse speckle patterns with equivalent $F_{LB}$ (see the $F_{LB}$ bands the figure). Figure 3b shows the integrated intensity (see Supplementary Fig. 5) as a function of $F_{LB}$ and indicates a maximum for $F_{LB} = 0.625$ (see magenta bar in the figure) that corresponds to a splitting ratio $r_{sp} = F_{LB}/(1 - F_{LB}) = 1.66$. The data from the static silica sample was used to determine the static contrast as a function of $F_{LB}$ (see Fig. 3c). This allowed us to determine the decoherence factor $\sigma_d$ for $F_{LB} = 0.625$. The red line in Fig. 3c shows a fit of Eq. (3) to the data and yields the decoherence factor $\sigma_d = 0.79$ and $\beta_0 = 0.29$ (see Methods).

**Split-pulse speckle contrast revealing nanosecond dynamics.** Using the methods and baseline parameters described above, the dynamics of a dilute suspension of 1-nm-radius gold nanoparticles in hexane were measured covering a range of wave vector $0.2 \text{ nm}^{-1} < q < 1.5 \text{ nm}^{-1}$. We accomplished this by determining the contrast of split-pulse speckle patterns measured with a fixed time delay $\Delta_t = 1.3$ ns and splitting ratio $r_{sp} = 1.66$ ($F_{LB} = 0.625$) as a function of the momentum transfer $q$ where the

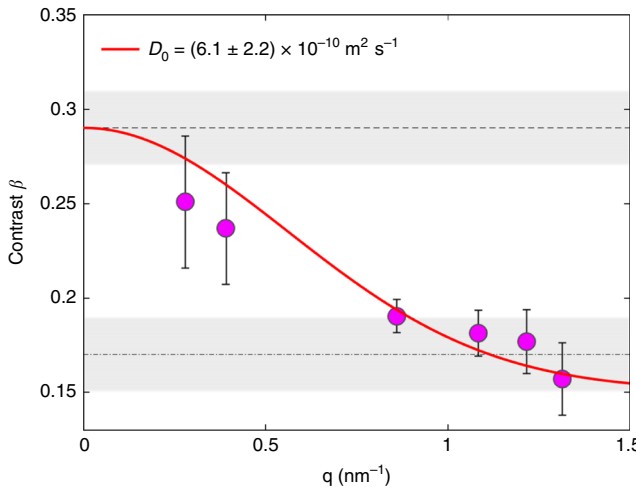

**Fig. 4** Speckle contrast revealing nanosecond dynamics. $q$ dependent contrast decay caused by diffusing gold nanoparticles and measured via the split-pulse XPCS method with two X-ray pulses separated by 1.3 ns. The red line corresponds to the model described in the text and yields the free particle diffusion coefficient $D_0$. The horizontal gray bands refer to the expected limiting contrast values for a splitting ratio of 1.66 ($F_{LB} = 0.625$). The center of each band is denoted by a horizontal line. The error of data points was calculated based on Supplementary Eq. (3) (see Supplementary Note 1)

relaxation times $\tau_c = 1/(D_0 q^2)$ spans from above to below $\Delta_t$. The contrast data shown in Fig. 4 reveal contrast values of about 0.25 at low $q$ ($<0.3 \text{ nm}^{-1}$) that fall off to about 0.15 at larger $q$ ($>1.2 \text{ nm}^{-1}$), as expected from the underlying sample dynamics (see Supplementary Note 4).

The $q$ dependence of the contrast is given by[35,39]

$$\beta(q, \Delta_t) = \beta_0 \left( \frac{r_{sp}^2 + 1 + 2r_{sp}(\exp(-D_0 q^2 \Delta_t))^2}{r_{sp}^2 + 1 + 2r_{sp}} \right) \quad (2)$$

where $\beta_0 = 0.29 \pm 0.02$ is the contrast value obtained from the static calibration sample measurements and $D_0$ is the single particle diffusion coefficient (see Methods). The solid line in Fig. 4 is a fit of Eq. (2) to the data yielding the diffusion coefficient $D_0 = (6.1 \pm 2.2) \times 10^{-10} \text{ m}^2 \text{ s}^{-1}$ which is in excellent agreement with the calculated self diffusion coefficient $D_0$ of 1-nm-radius gold nanoparticles in hexane (see Methods).

## Discussion

In conclusion, we present the results of the ultrafast hard X-ray photon correlation spectroscopy experiment carried out on a dynamic material on few-nanosecond time scales. Central to this approach is the measurement and analysis of low-count-rate integrated split-pulse speckle patterns generated via a pulse split-and-delay system at LCLS. Building on this work, expected gains in photon flux and in beam-splitting optics will enable study of weakly scattering atomic systems at LCLS. Split-and-delay efforts are also underway at SACLA (Japan)[40] and the European XFEL (Germany)[41], extending in particular the energy tunability of the optical system. Furthermore, the delivery of double X-ray pulses by the FEL itself has recently been demonstrated[42–44] and applied in the soft X-ray regime[45]. However, these approaches provide only a limited range of time delays of order femtoseconds or discrete steps of hundreds of picoseconds. Thus, optics-based split-and-delay systems such as the one demonstrated in this work are necessary to fill a critical time regime in the characterization of dynamic systems. This capability promises to

elucidate the underlying dynamics of a wide variety of systems and will enable the study of many physical processes therein.

## Methods

**Sample preparation.** The 1-nm-radius alkanethiol stabilized and dried gold nanoparticles NanoXact were purchased from NanoComposix. The particles were dissolved in hexane. After the preparation, the samples were filled in capillaries and sealed.

**Experimental setup.** The experiment was carried out using the setup shown in Fig. 1. Monochromatized ($\Delta E/E = 1.24 \times 10^{-4}$) LCLS pulses of energy $E = 7.9$ keV were first split in the vertical plane by a beamsplitter crystal (SP1) into two pulses that subsequently propagate via the reflection from thick Bragg reflectors (R1–R6) along two rectangular paths of different pathlength, the upper branch (UB) and the lower branch (LB), respectively. The pulses are recombined at the beam-mixer position SP2 and propagate co-linearly towards the sample. The beam splitting and mixing was accomplished with thin Si(422) Bragg crystals (thickness < 12 μm, $\Delta E/E = 1.47 \times 10^{-5}$)[40]. The time delay $\Delta_t$ between two split pulses can be varied between 0 and 2.66 ns. A compound refractive lens (CRL) was used downstream of the split-and-delay instrument to focus the beam to a diameter of about 16 μm at the sample. The photon flux at the sample positions was about $5 \times 10^7$ photons/pulse, limited by the transmission of the XCS instrument[37] ($2.57 \times 10^{-3}$) and the throughput of the pulse split-and-delay Si(422) optics ($3.6 \times 10^{-2}$)[28].

**Static speckle contrast analysis.** The static contrast $\beta_0$ as a function of $r_{sp} \equiv r_{LB}/(1 - F_{LB})$ is shown in Fig. 3c, averaged over equal-$q$ detector annuli from 0.15 to 0.26 nm$^{-1}$. The data are modeled by[35,39]

$$\beta_0(F_{LB}) = \beta_{SB} \left( \frac{r_{sp}^2 + 1 + 2\sigma_d r_{sp}}{r_{sp}^2 + 1 + 2r_{sp}} \right), \qquad (3)$$

where $\beta_{SB} = 0.32$ is the single branch contrast (see Supplementary Note 3) and $\sigma_d$ is a measure of the decoherence between the two beams at the sample position (see Supplementary Fig. 7). Perfect overlap $\sigma_d = 1$ would yield $\beta_0 = \beta_{SB}$ independent of the splitting ratio $r_{sp}$ as shown by the dashed line in Fig. 3c. The prediction for zero overlap is shown by the dashed-dot line. A single parameter fit of Eq. (3) yields a decoherence factor $\sigma_d = 0.79$. As shown in Fig. 3c the error bars are lowest for $F_{LB} = 0.625$ corresponding to the highest integrated intensity (Fig. 3b)

**Dynamical speckle contrast analysis.** The dynamic sample, a dilute suspension of 1-nm-radius gold nanoparticles dispersed in hexane, was measured at room temperature with the detector centered at two angles ($2\theta = 0°$ and $2\theta = 1.5°$). This permitted momentum transfers $q$ between 0.2 and 1.5 nm$^{-1}$ to be covered. The azimuthally averaged scattering intensity of $5 \times 10^4$ split pulse frames is shown in Fig. 2d (red circles). The intensity profile was modeled with the form factor for spherical particles and yielded a radius of 1.05 nm with a polydispersity of $\Delta R/R = 25\%$ in good agreement with the nominal values. Inter-particle interactions are not expected for a volume fraction of 0.5% and are in fact absent in the experimental data.

The contrast dependence on the sample dynamics is given by Eq. (2) where $\beta_0$ is the contrast at $q = 0$. The free particle diffusion coefficient for particles with a radius $R$ is

$$D_0 = \frac{k_B T}{6\pi\eta R} \qquad (4)$$

where $k_B$ is Boltzmann's constant and $\eta$ is the viscosity of hexane at temperature $T$. For a dilute suspension of 1-nm-radius gold nanoparticles dispersed in hexane ($\eta = 0.297$ mPa·s) at room temperature ($T = 295$ K), the diffusion coefficient is $D_0 = 7.3 \times 10^{-10}$ m$^2$ s$^{-1}$.

**Data availability.** Data supporting the findings of this study are available from the corresponding author on reasonable request.

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

## Acknowledgements

We acknowledge the support of the Hamburg Centre for Ultrafast Imaging (CUI). Use of the Linac Coherent Light Source (LCLS), SLAC National Accelerator Laboratory, is supported by the U.S. Department of Energy, Office of Science, Office of Basic Energy Sciences under Contract No. DE-AC02-76SF00515. We thank the XCS staff for technical support. We acknowledge financial support by DFG within SFB 925. We would also like to acknowledge the LCLS detector group for the help with operating the pnCCD detector. S.L. was supported by the National Research Foundation of Korea (NRF-2016K1A3A7A09005386). Work at Argonne National Laboratory was supported by the U.S. Department of Energy, Office of Science, Basic Energy Sciences, Materials Sciences and Engineering Division.

## Author contributions

W.R. and G.G. designed research. M.Si., S.S. and A.R. prepared and supervised the use of the XCS beamline procedures. W.R., F.L., SH, M.Su., A.R., S.S., M.Si., M.W., S.L., T.O., P. F., G.B.S. and G.G. performed the experiment. F.L. prepared the sample. W.R., S.H. and M.Su. analyzed and modeled the data. H.S.S. and T.O. provided split-and-delay optics. R. H. and L.S. helped with detector data corrections. W.R., S.H. and G.G. wrote the manuscript text. All authors reviewed and discussed the analysis and manuscript.
