## [Peer Review File · Nature Communications]

Reviewers' comments:

Reviewer #1 (Remarks to the Author):

This MS reports the first successful demonstration of an XPCS experiment carried with a split pulse and delay line system, so that the time scale measured was of the order of nanoseconds. While the actual physics information obtained was rather trivial (measurement of the diffusion coefficient of Au nanoparticles exhibiting Brownian motion in solution), its importance lies in the fact that it is in fact a benchmark experiment of this kind, complete with a fairly detailed approach to the statistical analysis of the split-beam speckle patterns (which can be only appreciated by reading the supplementary material and which appears to be correct), and will be influential for what we might expect to be many more similar studies of this type in the future. It should be published in Nature Communications.

Reviewer #2 (Remarks to the Author):

XPCS is one of the methods that is really gaining a lot at XFELs that enable exploring fast dynamics of disordered matter and transient states of complex matter. The authors are among the pioneers who have developed split-delay optical set-up for providing proper methodology. Considering the rather weak speckle intensities in a single split shot the success of such studies is highly depended on implementation of reliable processing approaches accounting for all contributions, including complex statistical analysis of the possible single shot jittering as well. This paper reports very important results demonstrating the great potential of split-pulse set-up by implementing of proper data evaluation. The authors, having found a correct way to analyze the data, were able to obtain reliable information from the correlation spectra evaluating the correct numbers for the Brownian motion of Au particles. After minor revision, the paper deserves publication since it provides the fast growing community using XFELs an excellent methodology to be used for experiments in broad research fields. Together with the supplementary information the paper contains all details that should become available and help the XFEL users. The following point needs to be clarified before publication:

1. How good the detector discriminates from 1, 2, 3 photons per pixels. Some graph (histogram) showing a clear statistics will be helpful.
2. Some comments on expectations and limits of using split-delay beams for monitoring dynamics in systems induced by various external stimuli will be appreciable.

Reviewer #3 (Remarks to the Author):

This is a very technical paper about the efforts to extend dynamic x-ray scattering (X-PCS) to the ultrafast regime at FEL sources. A beam splitter instrument was developed to this end and the paper reports first data of ns correlations taken at the SLAC FEL. I want to commend the authors for their effort which must have been tremendous. This is not coming through very strongly in the manuscript but given that the instrument was developed and first published 8 years ago and since first data are only available now, there must have been almost unsurmountable difficulties operating the device at a FEL.

However, the scientific content of the manuscript is, despite the heroic efforts, relatively thin. In essence, the two data points to the very left of figure 4 rising above the baseline are the only

indications of a signal that is shown here. I have strong doubts whether this is enough to warrant publication in Nature Communications, also because the sample is a colloidal gold suspension made for calibration purposes and hence of no scientific interest.

A few questions and remarks are listed below that I believe will help the authors improving the manuscript, possibly for submission to a more suitable journal where the tricky data analysis also can be highlighted better than in an appendix supplement.

1. Applications to study non-equilibrium processes are mentioned. However, the time-averaged procedure employed in the manuscript will mostly disable this option.
2. In eq. (1) Γ denotes the gamma function which is not the same as the gamma distribution (mentioned line 72).
3. Line 74: what does the \wedge sign over λ refer to?
4. I can see a strong correlation between the scattered signal and a monitor placed in the lower branch in Fig 3a. However, it is not obvious why this justifies use of the scattered signal as global monitor (I_0) for the strength of each pulse. Such an I_0 monitor would usually be placed before the splitter to be independent on for instance sample properties, CCD properties, etc. This is a possible source of systematic errors in the method employed here to calculate F_{LB} and r_{sp} .
5. Figure 3a is quite mysterious: I think one reason for the position of the F_{LB} contour lines is due to the postulated functional form of the $F_{LB}=1$ boundary (dashed line). It is mentioned that this line reflects the nonlinearity of the monitor but no details are given. Usual detector nonlinearity due to deadtime would to the best of my understanding lead to a different behavior so it is important to be much more precise here. Why is the dashed line drawn like shown in figure 3a and is there a solid reasoning behind?
6. One consequence of the above choice is the section of fig 3a into the shown F_{LB} bands. I don't understand the behavior of the scattering patterns as a function of F_{LB} , for example as expressed in Figure 6 (supplement) or in Fig. 3a (inset). Is there for instance a good explanation why $F_{LB}=0.85$ gives much stronger scattering patterns on the average? This seems counterintuitive and could question the entire procedure of determining F_{LB} (point 4 and 5 above). This strange unbalance in F_{LB} and F_{UB} values somehow point towards that the two branches behave quite differently. Is this due to the thin crystal (novel, line 139?) reflections only implemented in the upper branch?
7. Figure 3b: It seems strange that the static contrast determined is mostly above the red line (supposed to be a best fit) and in some cases even significantly above the postulated limiting value for the maximum contrast assuming perfect overlap and no decoherence. So is it the maximum possible contrast or not? This inconsistency could indicate that Poisson noise is mistakenly assigned to speckle contrast because negative binomial fitting to the data is not accurate.
8. Figure 4 (typo caption: depended): This is the main result of the paper. Why is the red model line allowed to dive below the minimum contrast line for $q \rightarrow \infty$? From this (and point 7 above) the reader gets the impression that the determined contrast values are highly uncertain and possibly wrong. Maybe this is somehow related to point 9:
9. Is the β_0 parameter used in eq. 2 q independent? Usually, this is not the case. I'm also wondering if this static contrast shouldn't be determined with the same sample as used for the dynamics investigations but in an arrested state. This might be difficult in practice but at least the

thicknesses should be comparable.

10. I'm also not convinced about the rescaled single branch speckle contrast (0.32) determined in the supplement. The speckle size is calculated for the purpose of scaling but the speckle size formula is very rough for instance not taking any shape factor into account.

11. Typo: Δt should be in ns (line 103).

12. In figure 5 (supplement) I'm not sure why the curve is flattening out at high F_{LB} and why no data points are shown for small F_{LB} ? It's like there are no weak pulses measured in any of the branches...

Reviewer #1 (Remarks to the Author)

This MS reports the first successful demonstration of an XPCS experiment carried with a split pulse and delay line system, so that the time scale measured was of the order of nanoseconds. While the actual physics information obtained was rather trivial (measurement of the diffusion coefficient of Au nanoparticles exhibiting Brownian motion in solution), its importance lies in the fact that it is in fact a benchmark experiment of this kind, complete with a fairly detailed approach to the statistical analysis of the split beam speckle patterns (which can be only appreciated by reading the supplementary material and which appears to be correct), and will be influential for what we might expect to be many more similar studies of this type in the future. It should be published in Nature Communications.

Response to the Reviewer 1:

We thank the Reviewer for his enthusiasm and appreciation towards our work. We agree with the reviewer's remark on the physics information but appreciate his conclusion "*its importance lies in the fact that it is in fact a benchmark experiment of this kind*".

Reviewer #2 (Remarks to the Author)

XCPS is one of the methods that is really gaining a lot at XFELs that enable exploring fast dynamics of disordered matter and transient states of complex matter. The authors are among the pioneers who have developed split-delay optical set-up for providing proper methodology. Considering the rather weak speckle intensities in a single split shot the success of such studies is highly depended on implementation of reliable processing approaches accounting for all contributions, including complex statistical analysis of the possible single shot jittering as well. This paper reports very important results demonstrating the great potential of split-pulse set-up by implementing of proper data evaluation. The authors, having found a correct way to analyze the data, were able to obtain reliable information from the correlation spectra evaluating the correct numbers for the Brownian motion of Au particles. After minor revision, the paper deserves publication since it provides the fast growing community using XFELs an excellent methodology to be used for experiments in broad research fields. Together with the supplementary information the paper contains all details that should become available and help the XFEL users. The following point needs to be clarified before publication:

Response to the Reviewer 2:

We very much appreciate the Reviewer's useful and positive comments on our work. The comments stimulated us to improve our manuscript. We provide below the answers.

Comment 1:

"How good the detector discriminates from 1, 2, 3 photons per pixels. Some graph (histogram) showing a clear statistics will be helpful."

The droplet algorithm procedure was successfully applied for analyzing single-pulse based data sets taken with the Princeton Instruments direct illumination CCD, with details elaborated in Hruszkewycz, S. O. *et al.* "High Contrast X-ray Speckle from Atomic-Scale Order in Liquids and Glasses." *Phys. Rev. Lett.* **109** (2012). In our work we applied the same algorithm to the data collected by the pnCCD detector. The efficiency of the algorithm and hence the discrimination between single, double and triple photon events is shown by the histogram of analog to digital units (ADUs) presented in Supplementary Fig. S1. The resulting histogram shows that the detector can distinguish between 1,2,3 photon events with a high sensitivity of 12σ . This value was determined from the ratio between the separation between the 1 and 2-photon peak $\Delta P_{1,2}$ to the standard deviation σ of the 1-photon peak.

We thank the Reviewer for this comment. We have updated the supplementary material with the additional information on how well the detection scheme can discriminate between single, double and triple photon events by updating the Figure S1 (Supplementary Material).

In line 29 (in the Supplementary Material) we have also added the sentence:

"The noise of the detector was derived from the σ value of the first photon peak and corresponds to 208 ADUs. The histogram shows that the detector can distinguish between 1,2,3 photon events with a high sensitivity of 12σ ".

Comment 2:

"Some comments on expectations and limits of using split-delay beams for monitoring dynamics in systems induced by various external stimuli will be appreciable."

Slow non-equilibrium dynamics e.g., in metallic glasses or alloys (after a fast temperature quench) has

been studied by XPCS at storage ring sources before (A. Malik et al., *Phys. Rev. Lett.* **81**, 5832 (1998); A. Fluerașu et al., *Phys. Rev. Lett.* **94**, 055501 (2005)). In these experiments non-equilibrium dynamics evolves on time scales very long (minutes) compared to the repetition rate (MHz) of the X-ray beam pulses, thus allowing the computation of two-time correlation functions with sufficient statistics at all relevant lag times without the need for multiple pump (or external stimulus) events.

Here we present a method that enables measurement of ultrafast non-equilibrium processes such as e.g. ultrafast demagnetization (Pfau et al., *Nature Communication* 3, 1100, 2012) that evolves after a fast (sub 100fs) IR pump-pulse on sub-ps timescales with subsequent re-magnetization on sub-ns timescales. The pump-induced dynamics in such a system has thus completely recovered when the subsequent pump - probe event occurs after 8.3 ms (120 Hz in the case for LCLS). Typical pump-probe experiments on such systems are usually carried out by integrating 1000 or more pump-probe events (Pfau et al., *Nature Communication* 3, 1100, 2012). This is of course only possible if the recorded process is reversible. In order to access the fluctuations accompanying e.g. demagnetization process one would carry out a pump - (split-pulse) probe experiment (as shown in the figure below) with a repetition rate considerable lower than the recovery time of the system, which is fulfilled by the mentioned system.

Then the number of pump - (split-pulse) probe events could be chosen at will in order to either increase statistics or in order to map out non-equilibrium dynamics as a function of the time ΔT between the pump and (split-pulse) probe pulses or the (split-pulse) delay time Δt . Similar considerations hold for photo-induced phonon dynamics or photo-induced phase transitions with eventually different timescales for the recovery of the process and thus different conditions for the sampling rate. Non-equilibrium dynamics is thus accessible to split-and-delay XPCS if the investigated process is reversible on the timescale set by the repetition rate of the pump - probe protocol.

We have introduced a clarifying sentence in line 26 of the manuscript:

"Such pump - (split-pulse) probe experiments are feasible for pump induced non-equilibrium processes that are reversible on the timescale defined by the repetition rate of the experiment".

Reviewer #3 (Remarks to the Author)

Response to the Reviewer 3:

We highly appreciate the Reviewer's useful comments and criticism of our work. The comments indicated to us that the Supplementary Material should provide a more comprehensive description of the impact of the random (spiky) nature of SASE free electron source, especially with regard to intensity monitors utilized in this experiment. To answer the questions and comments raised by the Reviewer, we have revised our manuscript as follows:

- i. We provide answers to the Reviewer's comments and questions. Since the Reviewers' comments and criticism are quite lengthy, we first provide answers to the general remarks followed by detailed answers to more specific questions.
- ii. We have removed figures 4 and 5 in the Supplementary Material and provided a new figure 4 with an extended explanation (lines 54 - 132) supporting our model for determining the splitting ratios.
- iii. We have updated figure 6 in the Supplementary Material (now figure 5).
- iv. We have updated figure 4 in the Manuscript and took into account the spread of the minimum and maximum contrast due to the fluctuations of the FEL beam size on a shot to shot basis.

Response to general comments:

Comment 1:

"This is a very technical paper about the efforts to extend dynamic x-ray scattering (X-PCS) to the ultrafast regime at FEL sources. A beam splitter instrument was developed to this end and the paper reports first data of ns correlations taken at the SLAC FEL. I want to commend the authors for their effort which must have been tremendous. This is not coming through very strongly in the manuscript but given that the instrument was developed and first published 8 years ago and since first data are only available now, there must have been almost unsurmountable difficulties operating the device at a FEL."

SASE X-ray free electron laser (XFEL) sources are the latest premier light sources with a number of challenging properties (short pulses, pointing instabilities, intensity fluctuations, random mode structure, etc). It is not surprising that many technical issues have to be solved before new scientific questions can be addressed. This is realized by many journals including the Nature family, where **many of such technical advances are published** (e.g., Riedel R. et al., "Single-shot pulse duration monitor for extreme ultraviolet and X-ray free-electron lasers", Nat. Comm. **4**, 1731 (2013), Ideguchi T. et al, " Adaptive real-time dual-comb spectroscopy", **5**, 3375 (2014), Schutz S. et al, " Femtosecond all-optical synchronization of an X-ray free-electron laser ", **6**, 5938 (2015), Grguras I. et al., "Ultrafast X-ray pulse characterization at free-electron lasers", Nat. Photon. **6** 852 (2012)). In our case we address the lack of a technique to probe ultrafast atomic-scale dynamics in spatially and temporally ensemble averaged systems such as liquids and glasses. Today, the X-ray split-and-delay approach is the only viable way to study such phenomena. And thus, multiple institutions in various facilities are developing similar types of devices, which will allow measuring and analyzing ultra fast dynamics from X-ray speckle data of which the foundations are demonstrated in our manuscript.

Through sustained effort and steady process since the first publication of this long-term project (Roseker, W. et al. "Performance of a picosecond x-ray delay line unit at 8.39 keV", Opt. Lett. **34**, 1768–1770 (2009)), we have reached the point where ultrafast XPCS is now viable and poised to make new discoveries in

the dynamics of matter as FEL sources continue to improve. However there were also further periodic reports on the progress in 2011 (Roseker W. et al., J. Synchr. Rad. **18**, 481 (2011)) and after installation at LCLS in 2012 (Roseker W. et al., Proc. SPIE 8504, X-Ray Free-Electron Lasers: Beam Diagnostics, Beamline Instrumentation, and Applications, 85040I (2012); doi:10.1117/12.929759). We note that the commissioning of the device was carried out on the basis of peer reviewed proposals and access to LCLS beam was thus possible twice a year for typically 5x12 hour shifts. It is nevertheless true that several difficulties had to be surmounted:

- i. Improved ultrathin perfect crystals were developed in parallel at Osaka University (Ref.27) and tested at the Spring8 facility (T. Osaka, et al, Proc. of SPIE Vol. 8848 884804-1: Optical Advances in X-Ray/EUV Optics and Components VIII (2013).) before installation in the device.
- ii. A new strain free mount had to be developed and tested for these crystals.
- iii. The optics at the XCS station had to be re-characterized since the beryllium focusing lenses were found to introduce astigmatism in the beam.
- iv. The beamline delivered less coherent flux to the sample than initially expected (Ref. 35) due to performance of beamline optics at the time of the measurement.

Issues i - iii have been successfully solved in collaboration with SLAC (USA), SACLA (Japan), ANL(USA), and DESY(Germany). Improvements to the XCS instrument are ongoing, addressing point iv. The work presented in this paper is the first demonstration of the viable split pulse XPCS for the measurements of atomic scale dynamics in materials at nanosecond time scales.

Comment 2:

“However, the scientific content of the manuscript is, despite the heroic efforts, relatively thin. In essence, the two data points to the very left of figure 4 rising above the baseline are the only indications of a signal that is shown here. I have strong doubts whether this is enough to warrant publication in Nature Communications, also because the sample is a colloidal gold suspension made for calibration purposes and hence of no scientific interest.”

Here we disagree strongly with the Reviewer’s conclusions:

- i. It is true that the contrast function is built of 6 data points with corresponding error bars. In addition we determined the limiting values of the contrast (i.e., $\langle \beta(Q \rightarrow 0) \rangle$, $\langle \beta(Q \rightarrow \infty) \rangle$) from the independent static sample measurements. The exponential form of the correlation function is undisputable and the result for the diffusion coefficient is the expected one. This proves the validity of the analysis. We note in that context that the very first X-ray Photon Correlation Spectroscopy publication at a storage ring (Brauer et al., “X-ray Intensity Fluctuation Spectroscopy Observations of Critical Dynamics in Fe₃Al,” Physical Review Letters **74**, 2919 (1995)) was based on data with a contrast level of about 3×10^{-3} !
- ii. It is evident that by remedying the limited beamline transmission and by eventually improving the transmission of the device and thus increasing the available flux at the sample (e.g., using lower index reflections) it will be possible to address the important systems mentioned in the introduction of the manuscript [Refs. 9-23]. These (mostly atomic scale disordered) systems will likely yield speckle patterns of comparable intensity to the ones described in the manuscript. The data will thus need to be analyzed with exactly the same maximum likelihood fitting procedures derived and communicated in this paper.
- iii. Verifying the validity of the Stokes-Einstein relation on ns-ps timescales is by no means an irrelevant thing to do. We mention in the motivation of the manuscript that one prime application of split-and-delay XPCS will be the study of fluctuations of water in the undercooled state [Refs. 11&12] where the Stokes-Einstein relation has been reported to be violated [see e.g. Sow-Hsin Chen et al., PNAS, 12974, 103/35 (2006)]. Simulations show that the relevant dynamics in the undercooled state are actually in the ns regime.

Detailed answers to the questions: Question 1:

"Applications to study non-equilibrium processes are mentioned. However, the timeaveraged procedure employed in the manuscript will mostly disable this option."

Slow non-equilibrium dynamics e.g., in metallic glasses or alloys (after a fast temperature quench) has been studied by XPCS at storage ring sources before (A. Malik et al., *Phys. Rev. Lett.* **81**, 5832 (1998); A. Fluerasu et al., *Phys. Rev. Lett.* **94**, 055501 (2005)). In these experiments non-equilibrium dynamics evolves on time scales very long (minutes) compared to the repetition rate (MHz) of the X-ray beam pulses, thus allowing the computation of two-time correlation functions with sufficient statistics at all relevant lag times without the need for multiple pump (or external stimulus) events.

Here we present a method that enables measurement of ultrafast non-equilibrium processes such as e.g. ultrafast demagnetization (Pfau et al., *Nature Communication* 3, 11, 2012) that evolves after a fast (sub 100fs) IR pump-pulse on sub-ps timescales with subsequent re-magnetization on sub-ns timescales. The pump-induced dynamics in such a system has thus completely recovered when subsequent pump -probe event occurs after 8.3 ms (120 Hz in the case for LCLS). Typical pump-probe experiments on such systems are usually carried out by integrating 1000 or more pump-probe events (Pfau et al., *Nature Communication* 3, 11, 2012). This is of course only possible if the recorded process is reversible. In order to access the fluctuations accompanying the demagnetization process one would carry out a pump - (split- pulse) probe experiment (as shown in the figure below) with a repetition rate considerable lower than the recovery time of the system, which is fulfilled by the mentioned system.

Then the number of pump - (split-pulse) probe events could be chosen at will in order to either increase statistics or in order to map out non-equilibrium dynamics as a function of the time ΔT between the pump and (split-pulse) probe pulses or the (split-pulse) delay time Δt . Similar considerations hold for photo- induced phonon dynamics or photo-induced phase transitions with eventually different timescales for the recovery of the process and thus different conditions for the sampling rate. Non-equilibrium dynamics is thus accessible to split-and-delay XPCS if the investigated process is reversible on the timescale set by the repetition rate of the pump - probe protocol.

We have introduced a clarifying sentence in line 26 of the manuscript:

"Such pump - (split-pulse) probe experiments are feasible for pump induced non-equilibrium processes that are reversible on the timescale defined by the repetition rate of the experiment"

Question 2:

"In eq. (1) Γ denotes the gamma function, which is not the same as the gamma distribution (mentioned line 72)."

The Reviewer was correct to point out this typo, which we corrected in the text. Indeed, we used the gamma function, as the referee surmised.

Question 3:

"Line 74: what does the $\hat{\lambda}$ sign over λ refer to?"

The $\hat{\lambda}$ sign over λ denotes the observed *mean count rate* value. We thank the Reviewer for pointing this out. For clarity, we changed the following sentence (line 77 in revised version of the manuscript):

"The observed count rate ($\hat{\lambda}$)..."

to:

"The observed mean count rate $\hat{\lambda}_j$ (taken as an estimate of the true count rate) ..."

Question 4:

"I can see a strong correlation between the scattered signal and a monitor placed in the lower branch in Fig 3a. However, it is not obvious why this justifies use of the scattered signal as global monitor (I_0) for the strength of each pulse. Such an I_0 monitor would usually be placed before the splitter to be independent on for instance sample properties, CCD properties, etc. This is a possible source of systematic errors in the method employed here to calculate F_LB and r_sp."

The approach of using a monitor placed before the beamsplitter as a global monitor is commonly used with sources that provide uniform pulse spectrum (e.g., Storage Ring sources). At an FEL source, because of the spiky spectrum of the incident beam, it is however inappropriate to use an I_0 monitor upstream of a narrow-bandpass optical system such as the split-and-delay instrument described here. This is because the spectrum of an FEL source consists of randomly distributed spikes (as shown in Figure S4a). The beamsplitter of the split-and-delay instrument was set such that two different portions of the FEL spectrum were selected into the lower and upper branch, respectively. In such configuration, the correlation between I_0 and downstream monitors is weak. Therefore, an I_0 monitor cannot provide a proper normalization of pulse intensities. We have extended Section 2 in the Supplementary Material to provide more details on the operation setting of the split-and-delay and calculations of F_LB and r_sp using the pnCCD as a "global monitor". Furthermore we want to point out that the pnCCD as a "global" monitor is very accurate and sensitive detector for low count rates.

Question 5:

"Figure 3a is quite mysterious: I think one reason for the position of the F_LB contour lines is due to the postulated functional form of the F_LB=1 boundary (dashed line). It is mentioned that this line reflects the nonlinearity of the monitor but no details are given. Usual detector nonlinearity due to deadtime would to the best of my understanding lead to a different behavior so it is important to be much more precise here. Why is the dashed line drawn like shown in figure 3a and is there a solid reasoning behind?"

We thank the Reviewer for pointing out that we were not precise enough with our explanation regarding the non-linear behavior of the monitor. The observed non-linearity shown in Figure 3a is not related to the performance of the monitor (e.g., deadtime) but it is due to the different energy bandwidths from spiky energy spectrum of the FEL seen by the monitor and the pnCCD detector, respectively. We have rewritten the section 2 in the Supplementary Material and provided full details on:

- the non-linear correlation between the monitor and the detector
- the position and form of the $F_{LB} = 1$ boundary.

Our explanation is supported by the simulation of FEL pulses propagating through a single branch of the split-and-delay.

Question 6:

"One consequence of the above choice is the section of fig 3a into the shown F_LB bands. I don't understand the behavior of the scattering patterns as a function of F_LB, for example as expressed in Figure 6 (supplement) or in Fig. 3a (inset). Is there for instance a good explanation why F_LB=0.85 gives much stronger scattering patterns on the average? This seems counterintuitive and could question the entire procedure of determining F_LB (point 4 and 5 above). This strange unbalance in F_LB and F_UB values somehow point towards that the two branches behave quite differently. Is this due to the thin crystal (novel, line 139?) reflections only implemented in the upper branch?"

The question involves understanding the behavior of scattering patterns as a function of F_LB. As pointed out by the Reviewer the two branches indeed behave differently. This is a consequence of the operation mode of the split-and-delay splitter and the nature of FEL spectrum (explained in the response to the **question #4**). In this configuration the lower branch had a higher mean intensity compared to the upper branch. Therefore, the F_LB bands higher than 0.5 are expected to have higher mean intensities. We have provided a detailed explanation regarding the intensity distribution of the split-pulse pattern intensities as a function of F_LB in lines 121 - 132 of the Supplementary Material. We have also modified Figure S6

(now Figure S5) in the Supplementary Material by adding the integrated intensity plot vs F_LB, which should provide more clarity to our explanation.

As suggested by the Reviewer we removed the word "novel" in line 139.

Question 7:

"Figure 3b: It seems strange that the static contrast determined is mostly above the red line (supposed to be a best fit) and in some cases even significantly above the postulated limiting value for the maximum contrast assuming perfect overlap and no decoherence. So is it the maximum possible contrast or not? This inconsistency could indicate that Poisson noise is mistakenly assigned to speckle contrast because negative binomial fitting to the data is not accurate."

The Reviewer asks why the static contrast, shown in Fig. 3b, is mostly above the red line and also why the contrast corresponding to the full overlap case presented by the dashed is the maximum contrast. The maximum contrast value denoted by the dashed line in Fig. 3b is determined (as given in eq. 3) by the single branch contrast β_{γ}^* . As we noted in the response to the **question #8**, the value of β_{γ}^* can vary due to FEL beamsize fluctuations that lead to speckle size fluctuations. Therefore, the value of β_{γ}^* can vary and we accepted a 4% contrast span that experimentally accounts for the aforementioned fluctuations. This value is also reflected by the error bar of the static contrast measured at F_LB = 0.625. In order to address this point, we changed the word "maximum contrast" to "maximum expected contrast" (caption of Fig. 3, caption of Fig.4). We would also like to note that only two contrast points (out of 10), shown in Figure 3b, are above the maximum expected contrast. The other eight points (see the errorbars) are within the expected static contrast boundaries.

The Poisson noise was not assigned mistakenly to a speckle contrast since the negative binomial fitting takes into account Poisson noise contributions.

Question 8:

"Figure 4 (typo caption: depended): This is the main result of the paper. Why is the red model line allowed to dive below the minimum contrast line for $q \rightarrow \infty$? From this (and point 7 above) the reader gets the impression that the determined contrast values are highly uncertain and possibly wrong. Maybe this is somehow related to point 9:"

Figure 4(in the manuscript) shows the speckle contrast as a function of wave vector transfer. The red line corresponds to the model of 1nm radius Au particles dispersed in hexane and it yields the free particle diffusion coefficient. At the higher q values the red line goes below the horizontal dashed-dot line. The dashed-dot line represents the expected minimum contrast in the experiment. However, the value of minimum contrast is sensitive to the single branch contrast β_{γ}^* and the change in β_{γ}^* from 0.32 to 0.30 will shift the minimum contrast from 0.17 to 0.15. The static contrast point measured at F_LB = 0.625 shows an errorbar of 4% which is taken as a measure of an expected contrast fluctuations.

As discussed in the response to the **question 7**, a determination of speckle contrast is highly affected by FEL shot to shot beam size fluctuations. In this case, a 4% static contrast variation is found as an acceptable error bar. However, we would like to note that our statement minimum contrast was too strong and we changed it in the manuscript to "expected minimum contrast". We also updated Figure 4 (in the manuscript) by including the spread of the expected maximum and minimum contrast (denoted by the grey bands).

We thank the Reviewer for finding the typo in word "**dependent**". We corrected the typo in the revised version of manuscript.

Question 9:

"Is the β_0 parameter used in eq. 2 q independent? Usually, this is not the case. I'm also wondering if this static contrast shouldn't be determined with the same sample as used for the dynamics investigations but in an arrested state. This might be difficult in practice but at least the thicknesses should be comparable."

The contrast of a static sample i.e, β_0 is q dependent and is reflected by the contrast decrease as a function of the scattering angle. The strength of this decrease depends on the experimental conditions (mostly the longitudinal coherence length). In a typical x-ray photon correlation spectroscopy (XPCS) experiment performed in a small angle x-ray scattering configuration (SAXS) with Si(111) monochromator, the contrast decrease is negligible, and therefore, can be considered as constant. In our experiment ($E = 7.9$ keV) the longitudinal coherence length was defined by the high index reflections (Si422) of the split-and-delay, which is 8 times higher compared to Si(111). This allows us to approximate that the β_0 is constant within the scattering angles accessed in the experiment (< 1.5 degrees). The dried silica static sample was contained in a capillary tube that was the same size as the one used to hold the dilute Au solution.

Question 10:

"I'm also not convinced about the rescaled single branch speckle contrast (0.32) determined in the supplement. The speckle size is calculated for the purpose of scaling but the speckle size formula is very rough for instance not taking any shape factor into account."

The single branch speckle contrast was calculated according the speckle formula given in the literature (Sandy, A. R. *et al. J. Synchr. Rad.* **6**, 1174–1184 (1999)). As pointed out by the Reviewer the formula does not take into account any shape factor. However, the speckle shape contribution comes from the cosine term of the scattering angle, i.e $\cos(2\theta)$ (described in detail Hruszkewycz, S. O. *et al. Phys. Rev. Lett.* **109** (2012)). Our experiment was performed in the small angle configuration with the largest angle 2θ of 1.5 degree. Therefore, speckle shape contributions are negligible.

Question 11:

"Typo: Δt should be in ns (line 103)."

We thank the Reviewer for finding this typo. We corrected it in the manuscript.

Question 12:

"In figure 5 (supplement) I'm not sure why the curve is flattening out at high F_{LB} and why no data points are shown for small F_{LB} ? It's like there are no weak pulses measured in any of the branches..."

We greatly thank the Reviewer for finding it out. The curve in the figure is flattening out however the error bars are increasing due to lower statistics. We attached the updated plot showing all points below. Nevertheless, we removed the plot from the Supplementary Material since it did contribute to the explanation of splitting ratios.

Figure 1. Correlation of lower branch monitor fraction F_{LB} to the upper branch monitor fraction F_{UB} .

Reviewers' comments:

Reviewer #2 (Remarks to the Author):

The authors took into account and responded satisfactory to my suggestions and criticism. The Ms is revised accordingly and can be accepted for publication.

Reviewer #3 (Remarks to the Author):

The reply and corrections to the manuscript improved things but the outstanding and most important question is still whether figure 4 is trustworthy or not. Without this result being demonstrated beyond any reasonable doubt the manuscript is not describing a novel technique on a calibration standard. I'm very doubtful whether figure 4 stands beyond any reasonable doubt.

A detailed discussion with graphical documentation is missing where the reader can see the photon statistics material on which the contrast values of ~25% have been determined for the two points to the left in figure 4. I would like to see demonstrated that a contrast of 15% (like found for the four rightmost data points) is not fitting the data. This is the most difficult part I know, but if it isn't shown the decay is not documented imo.

Several observations make me uneasy here: the significantly larger error bars on the left data points (reasoning behind?), the fact that these points are taken in a different run and with a different sample-detector distance (and hence with a different resolution and speckle size, and maybe different settings?), and the fact that data appear to be selected and filtered quite strongly according to splitting ratio and beam parameters in general.

I also cannot get the photon statistics to make sense on the few data shown: In figure 2c there are 33 photons in 22^2 pixels so 0.07 ph/pixel/pattern. That is more than one order of magnitude off the values claimed in figure 2d and the statements made in the manuscript (0.002-0.004). Obviously, if there're such inconsistencies in the analysis there is a real problem.

Finally the manuscript is not addressing the fact that the contrast depends on q and will decrease with q , even for a static sample. I raised this point in my first report, particularly mentioning the problem with sparse data that can lead to varying contrast also in the SAXS regime when the intensity varies. The authors simply reply that fitting with the negative binomial distribution takes care of this. While this is of course correct in theory, please demonstrate that this is true for your photon statistics data on the static sample using a few q partitions.

Reviewer 3

We acknowledge the comments of the Reviewer and we answer the remaining.

Questions:

1. “A detailed discussion with graphical documentation is missing where the reader can see the photon statistics material on which the contrast values of ~25% have been determined for the two points to the left in figure 4. I would like to see demonstrated that a contrast of 15% (like found for the four rightmost data points) is not fitting the data. This is the most difficult part I know, but if it isn't shown the decay is not documented imo.”

The simplest approach for extracting contrast from a low intensity speckle pattern is by using the negative binomial distribution (see eq. (1) in the manuscript). For very low intensities the contrast can be estimated by the ratio R of double photon $P(2)$ to single photon $P(1)$ events . This ratio yields

$$R = \frac{2 \times P(2)[1-P(1)]}{P(1)^2} - 1 \quad (1)$$

The validity of this estimate was demonstrated successfully by Hruszkewycz et al., *PRL* **109** 185502 (2012).

This method depends on an accurate determination of double photon events and it becomes very difficult when the mean speckle pattern count rate falls to the level treated here especially due to the uncertainties introduced by very weak intensity pulses from the FEL. By selecting however only the strongest speckle patterns and using a single fraction $F_{LB} = 0.625$, such an analysis becomes possible. Figure 1 shows the probability of single photon $P(1)$ and double photon $P(2)$ events for a subset of very strong patterns and a splitting ratio $F_{LB} = 0.625$ as a function of Q .

The R ratio was calculated according to eq. 1 (above) and the change in the contrast estimation is shown in Figure 2. The result shows clearly and beyond any doubt that even the simplest contrast estimator R gives higher values at low Q above a lower contrast baseline at larger Q 's. This is independent proof of a “contrast” decrease vs Q due to the sample dynamics. However, this analysis produces errorbars that are considerable. This can be overcome by the maximum likelihood algorithm that allows to treat all frames (including very weak ones) leading to lower uncertainties of the measured contrast and the result is the one shown in the manuscript (in Figure 4).

Note that, we attached a new Figure S2 in Supplement Material showing a detailed graphical documentation on the maximum likelihood determination of the contrast data shown in Fig. 4. The two top plots of Figure S2 show of χ_{ML}^2 vs M collected at $Q = 0.279\text{nm}^{-1}$ and $Q = 0.391\text{nm}^{-1}$, respectively. The minima of these curves give $M=4\pm 0.3$ and 4.2 ± 0.3 and correspond to contrast values of 25 +/- 3% and 24 +/- 3 %, respectively. The lower four plots correspond to the data obtained at higher Q values ($Q > 0.08\text{nm}^{-1}$). It is clear from those 6 plots that a contrast of 17% corresponding to $M = 5.9$ is impossible to reconcile with the χ_{ML}^2 curves for the two low Q data sets and thus does not fit the low Q data.

In lines: 49 to 62 (in Supplementary Material) we provided more detailed description of our analysis: “The maximum likelihood approach allows photon statistics from patterns with very low count rates to be meaningfully integrated into the estimate of M of the entire data set. Figure S2 shows χ_{ML}^2 plots as a function of M for various q values and $F_{LB} = 0.625$ determined numerically from the data collected on the sample of 1-nm radius gold particles in hexane solution. The minimum value of χ_{ML}^2 provides an estimate of M , and the curvature about this point is a measure of the uncertainty. The data sets collected

at the low q values (0.279nm^{-1} and 0.391nm^{-1}) consist of 2.3×10^4 patterns. The high q value points (i.e., 0.86nm^{-1} , 1.084nm^{-1} , 1.217nm^{-1} and 1.315nm^{-1}) correspond to four data sets measured each with 3.8×10^4 speckle patterns. The M values shown in Fig. S2 vary from $M = 4 \pm 0.3$ at 0.279nm^{-1} to $M = 6.4 \pm 0.6$ at $q = 1.315\text{nm}^{-1}$. The corresponding contrast values 0.25 ± 0.03 , 0.24 ± 0.03 , 0.19 ± 0.01 , 0.18 ± 0.01 , 0.18 ± 0.02 and 0.16 ± 0.02 are plotted in Fig. 4. This approach allows for a global evaluation of an entire ensemble of patterns at selected splitting ratio and q value thus minimizing the uncertainty of M and the speckle contrast value. This is an essential advantage compared to the contrast evaluation of individual patterns with the help of negative binomial distribution method⁴ (Supplement).”

Figure 1. Probabilities of a single and double photon events per pixel and per frame for fraction $F_{LB} = 0.625 \pm 0.025$ and mean photon intensities higher than $2e-3$.

Figure 2. Change of the contrast estimator R as a function of Q for $F_{LB} = 0.625 \pm 0.025$.

2. “Several observations make me uneasy here: the significantly larger error bars on the left data points (reasoning behind?), the fact that these points are taken in a different run and with a different sample detector distance (and hence with a different resolution and speckle size, and maybe different settings?), and the fact that data appear to be selected and filtered quite strongly according to splitting ratio and beam parameters in general.”

The Reviewer is mistaken in this point. The “left data” (i.e., $2\theta = 0^\circ$) points were taken with the same settings and with **the identical sample - detector distance** as the rest of the data points (corresponding $2\theta = 1.5^\circ$). Therefore, the resolution and the speckle sizes are the same. Furthermore, all data points were taken at two successive back to back measurements within one shift (12 hours). First at $2\theta = 0^\circ$, then right after at $2\theta = 1.5^\circ$. No change in the FEL operation nor alignment of the split-and-delay system or XCS instrument were taking place.

The reason for the larger error bars of the low Q data points is simply due to the lower statistics compared to the rest of data points. As shown in equation S2 (in Supplementary Material) the number of speckle patterns and number of pixels per Q bin contribute to maximum likelihood statistics. The plots below (Figure 3) show the data for $F_{LB} = 0.625$ collected at $2\theta = 0^\circ$ consisting of 22790 patterns which is about half of the scattering patterns at the $2\theta = 1.5^\circ$. The number of pixels per Q bin at $2\theta = 0^\circ$ is also much smaller compared to the number of pixels per Q bin taken at $2\theta = 1.5^\circ$. This implies the larger error bars at the low Q values.

Figure 2(left) Number of pattern in the selected Q bins. (right) Number of pixels in the selected Q bins

3. “I also cannot get the photon statistics to make sense on the few data shown: In figure 2c there are 33 photons in 22^2 pixels so 0.07 ph/pixel/pattern. That is more than one order of magnitude of the values claimed in figure 2d and the statements made in the manuscript (0.002-0.004). Obviously, if there’re such inconsistencies in the analysis there is a real problem.”

The mean intensity of the selected ROI in the depicted FEL pattern (in Figure 2c) is 0.07 photons/pixel. In this example, we selected one of the strongest FEL speckle patterns for illustration rather than presenting the reader with the mean intensity pattern with only very few detected photons. We have made note of this in the caption of the Figure in question so as to avoid confusion. Because the SASE FEL intensity fluctuates on a shot to shot basis, the mean value of an ROI in any individual shot is not necessarily a good depiction of the mean intensity of the entire data set.

In line 64 we have changed the word “typical” with “strong”.

4. “Finally the manuscript is not addressing the fact that the contrast depends on q and will decrease with q, even for a static sample. I raised this point in my first report, particularly mentioning the problem with sparse data that can lead to varying contrast also in the SAXS regime when the intensity varies. The authors simply reply that fitting with the negative binomial distribution takes care of this. While this is of course correct in theory, please demonstrate that this is true for your photon statistics data on the static sample using a few q partitions.”

Figure 4. Integrated intensity as a function q . Vertical lines denote the selected q bins.

As the reviewer suggested, we did utilize a static sample to test the contrast determination method presented here, and we took care to ensure that the falloff of average intensity with q did not bias our results. In our analysis, the intensity data were divided into several q -bins such that the intensity in each bin did not vary by more than 1% (as shown in the figure 4 for static sample. 18 q -bins are depicted by vertical lines). For each q -bin a contrast value was determined by using the intensity within the bin. Therefore, the evaluated contrast cannot be affected by wrongly assigned mean intensity to the negative binomial distribution.

The uncertainty in the determination of the speckle contrast depends on how well single, double and triple photons events can be determined experimentally. The pnCCD detector used in this experiment was accurately characterized and tuned to the scattering intensities expected from the dynamical sample. In order to establish a correct and reliable contrast value from the static sample we intentionally chose the static sample (i.e., Silica particles) such that the scattering intensities at the detector matched the ones of the dynamic sample. This insured that we could measure the contrast for both (static and dynamic) samples with the same photon statistics for the low q values (as shown in the figure 2d).

Since the scattering intensity of the silica sample drops down very quickly as a function of q (actually q^{-4}) it is experimentally unfeasible with this sample to determine additional contrast values at higher q due to too low photon intensities.

We have already provided an answer to the question about the contrast dependence on the q value in our previous response to the Reviewer. In theory, the value of the static contrast is q dependent for large scattering angles (large q values) and low energy resolution of the beam. Our experimental conditions are quite opposite, namely we measured at very small scattering angles ($2\theta \leq 1.5^\circ$) and use very high energy resolution (one order of magnitude higher compared to typical SAXS experiments, $\Delta E/E = 1.4e-4$). Considering the aforementioned experimental conditions, the theoretical static contrast drop is less than 1% and therefore this effect is minor.

REVIEWERS' COMMENTS:

Reviewer #3 (Remarks to the Author):

I acknowledge the author response to my previous criticism. It appears that the decay observed in figure 4 is related to dynamics of the calibration standard. This is supported by the new information given in the supplementary material. However, the manuscript remains extremely technical with only limited or no interest for the reader interested in condensed matter dynamics. I also doubt that the specific hardware presented here will have any general impact on experimental physics. Hence, I stand by my original recommendation of submission to a more specialized journal.